# Dentin Mechanobiology: Bridging the Gap between Architecture and Function

**DOI:** 10.3390/ijms25115642

**Published:** 2024-05-22

**Authors:** Xiangting Fu, Hye Sung Kim

**Affiliations:** 1Institute of Tissue Regeneration Engineering (ITREN), Dankook University, Cheonan 31116, Republic of Korea; fuxiangting@dankook.ac.kr; 2Mechanobiology Dental Medicine Research Center, Cheonan 31116, Republic of Korea; 3Department of Nanobiomedical Science and BK21 NBM Global Research Center for Regenerative Medicine, Dankook University, Cheonan 31116, Republic of Korea

**Keywords:** dentin, mechanobiology, mechanosensing, mechanotransduction, viscoelastic properties, dentin-mimicking in vitro platforms

## Abstract

It is remarkable how teeth maintain their healthy condition under exceptionally high levels of mechanical loading. This suggests the presence of inherent mechanical adaptation mechanisms within their structure to counter constant stress. Dentin, situated between enamel and pulp, plays a crucial role in mechanically supporting tooth function. Its intermediate stiffness and viscoelastic properties, attributed to its mineralized, nanofibrous extracellular matrix, provide flexibility, strength, and rigidity, enabling it to withstand mechanical loading without fracturing. Moreover, dentin’s unique architectural features, such as odontoblast processes within dentinal tubules and spatial compartmentalization between odontoblasts in dentin and sensory neurons in pulp, contribute to a distinctive sensory perception of external stimuli while acting as a defensive barrier for the dentin-pulp complex. Since dentin’s architecture governs its functions in nociception and repair in response to mechanical stimuli, understanding dentin mechanobiology is crucial for developing treatments for pain management in dentin-associated diseases and dentin-pulp regeneration. This review discusses how dentin’s physical features regulate mechano-sensing, focusing on mechano-sensitive ion channels. Additionally, we explore advanced in vitro platforms that mimic dentin’s physical features, providing deeper insights into fundamental mechanobiological phenomena and laying the groundwork for effective mechano-therapeutic strategies for dentinal diseases.

## 1. Clinical Implications of Dentin Mechanobiology in Dental Diseases

According to the WHO Global Oral Health Status Report (2022), over 45% of the global population, amounting to 3.5 billion people, suffer from oral diseases such as dental caries, periodontitis, tooth loss, and oral cancers [1]. Additionally, dental diseases associated with aging are unavoidable, and there are no standardized guidelines for treating these conditions in older adults. Despite the significant impact on quality of life, dental diseases are not fatal, which has led to a lack of emphasis on dental research. For example, there are no effective treatments for dentin hypersensitivity because the underlying mechanisms are not well understood. Emerging evidence indicates that understanding dentin mechanobiology is essential for comprehending dental diseases, particularly those related to pain. Over the past two decades, there has been a gradual increase in research articles related to dentin mechanobiology, although the accumulated number of articles remains relatively small. This suggests that, despite its importance, this topic has not received significant attention. Therefore, this review aims to thoroughly examine how the physical properties of dentin interact with the unique mechano-sensory system of the tooth and how this knowledge can be leveraged to develop effective therapeutics for dental diseases.

### 1.1. Dentin and Its Mechanical Roles in Tooth Function

The tooth is continuously exposed to exceptionally high levels of mechanical loading, such as occlusal and masticatory forces, typically reaching up to approximately 1642.8 Newton, as a result of chewing or grinding [2]. Despite these extreme mechanical stresses, teeth exhibit remarkable resilience, maintaining their healthy condition. This resilience suggests the presence of inherent mechanical adaptation mechanisms within the tooth structure to counter constant mechanical stress. While bulk mechanical studies at the tissue level have been extensively conducted to understand the mechanical roles of dentin in tooth protection [3,4,5,6,7,8], mechano-sensing and -transduction at the cellular and molecular levels have not been comprehensively elucidated yet.

When examining tooth anatomy, it becomes evident that teeth comprise three distinct regions: enamel, dentin, and pulp [9]. Among these, dentin plays a pivotal role in mechano-protection and -sensing. Positioned between enamel and pulp, dentin exhibits softer and more viscoelastic properties compared to enamel, serving as a physical buffer by imparting flexibility to the tooth [10]. Consequently, teeth can withstand occlusal and masticatory forces without fracturing, thereby safeguarding the pulp tissues from infection and damage.

At the cellular level, odontoblasts, specialized cells residing within the dentin, function as mechano-sensors. Odontoblasts possess the capability to detect various external stimuli, including temperature, chemicals, and mechanical forces [11,12]. These odontoblasts transmit sensory signals to the nerves within the pulp, enabling the perception of external stimuli or pain and facilitating appropriate responses. In summary, dentin plays a crucial role in mechanically supporting human tooth function, thereby contributing to the maintenance of tooth health even under conditions of high mechanical loading. We will discuss the details of how dentin functions as mechano-sensors in Section 2.

### 1.2. Dentin-Associated Diseases

Dentin exposure can arise from various sources, including physical, chemical, pathological, biological challenges, or developmental abnormalities, leading to dental and periodontal damage, defects, and potential tooth loss [13,14,15]. Such exposure may occur due to enamel attrition [16], erosion [17,18], abrasion, abfraction [19], or periodontal tissue loss, including gingival recession, which exposes cervical and root dentin [20,21]. Aging and soft tissue dehiscence, including aggressive brushing, can also contribute to the apical displacement of gingival margins, leading to dentin exposure. Consequently, dentin exposure heightens sensitivity to external stimuli, often resulting in pain and discomfort, thereby diminishing the patient’s quality of life [21].

Dentin hypersensitivity (DHS) stands as one of the most common complaints among patients in dental clinics, with a prevalence ranging from 10% to 30% in the general population [22]. DHS is closely associated with dentin exposure, particularly the exposure of open dentinal tubules, and heightened dental pulp nerve responsiveness to external environmental stimuli [23]. Consequently, patients with DHS experience pain in response to typically non-harmful environmental stimuli such as gentle touch, mild cold or heat, acidic or sweat flavors, and air-flow stimuli, significantly impacting daily activities such as eating, drinking, speaking, and tooth brushing. In severe cases, DHS may lead to psychological and emotional distress and even chronic dental pain conditions [24,25]. Moreover, chronic dentin exposure further compromises the passive barrier role of dentin for the pulp, elevating the risk of pulp infection and eventual tooth loss. In clinical practice, advising patients about diet or toothbrushing has been a common approach to managing DHS [26]. Additionally, significant effort has been directed toward identifying and using appropriate materials for treatment, focusing primarily on sealing dentinal tubules or interfering with nerve impulse transmission [27]. Despite these efforts, current treatments for DHS remain only marginally effective.

In addition to structural damage or defects, alterations in the physical characteristics of dentin resulting from damage repair and aging significantly affect tooth function. Upon dentin damage or exposure, odontoblasts within dentin become activated and deposit a highly amorphous dentin matrix known as tertiary dentin to block the damage site. Such reparative reactions cannot fully restore the original dentinal tubular structures and permeability, thereby resulting in reduced dentinal sensitivity and weakened sensory responses [28,29]. Age-related senescence of dentin induces various alterations in physical characteristics, including increased hardness and elastic modulus in the mantle dentin [28,29]. Dentinal thickness increases with more mineral deposition, leading to a continual reduction in the pulp cavity. Simultaneously, reductions in the density of dentinal cells, including odontoblasts, sub-odontoblasts, and pulp fibroblasts, potentially diminish dentin sensitivity [30] and compromise dentin-pulp repair capacity [28]. Moreover, changes occur in the mineral composition of dentin, resulting in reduced density and increased porosity. These alterations may weaken its mechanical strength and resilience, rendering it more susceptible to damage and wear [31,32].

Despite dentin-associated diseases being recognized as clinically significant dental problems, the underlying pathogenesis mechanisms, particularly regarding abnormal sensitivity and pain transduction mechanisms, remain elusive. Our current understanding primarily relies on data obtained from in vitro and in situ studies and epidemiological surveys, as in vivo studies face limitations in sample accessibility [21]. For instance, while the hydrodynamic theory proposes fluid movements within exposed dentinal tubules as the underlying mechanism for DHS pain [18], it fails to account for all pain associated with DHS and does not necessarily lead to effective treatment. Similarly, there are no established guidelines for treating older patients or adjusting treatments based on the physical characteristics of dentin in clinical practice due to our limited understanding. Hence, there is a pressing need to investigate the exact pathogenesis mechanisms of dentin-associated diseases, especially how the physical characteristics of dentin affect sensory function and pain transduction, to facilitate a more comprehensive understanding and establish effective treatments.

## 2. Dentin Mechano-sensing

### 2.1. Key Physical Features of Dentin in Tooth Mechano-sensing

Dentin plays a pivotal role in human tooth function, offering crucial physical support, protection, and the ability to sense mechanical stimuli owing to its distinctive physical characteristics. Dentin is mineralized connective tissue primarily composed of 72% inorganic material, predominantly hydroxyapatite and non-crystalline amorphous calcium phosphates, along with 20% organic materials, including type I collagen (90%) and dentin-specific proteins (10%) such as dentin sialophosphoprotein (DSPP), dentin matrix protein-1 (DMP-1), osteopontin (OPN), and osteocalcin (OCN), as well as 8% water [33]. Unlike enamel, dentin exhibits less brittleness and more viscoelasticity, providing the necessary flexibility to withstand occlusal and masticatory forces without fracturing and safeguarding pulp tissues from microbial and other harmful stimuli [34,35]. This resilience is credited to the fibrous arrays of type I collagen, proving tensile strength, and to the mineralized crystals within these collagen fibers, offering high compressive strength and rigidity [35,36]. Meanwhile, oral bacteria such as streptococci, the primary bacterial colonizers of the oral cavity, exploit type I collagen fibrils within dentinal tubules for their invasion, both physically and chemically [37].

Structurally, dentin exhibits a hierarchical arrangement with micro-sized tubules known as dentinal tubules embedded in a nano-fibrous network of collagens and mineral crystals. These dentinal tubules, with diameters ranging from 0.2 to 0.5 µm and densities of 30,000–80,000 tubules/mm^2^, extend outward from the pulp, forming a three-dimensional network with collateral branches measuring 1 µm in diameter at specific angles, crisscrossing intertubular dentin [18,38]. This architecture allows for high permeability, enabling fluid flow through the dentinal tubules to facilitate molecular transportation and the detection of external stimuli through hydrodynamic changes.

Odontoblasts, specialized cells situated at the dentin-pulp interface, are crucial for dentin formation, sensing, and responsive/defensive function. Their cell bodies establish physical contact with nerve endings in the pulp, while their processes extend into dentinal tubules [39] (Figure 1). Due to the special anatomical location of odontoblasts in the tooth structure, odontoblasts are the first cellular component to encounter external stimuli or bacterial damage, indicating their potential role as sensors or immune cells. Through various types of receptors in their membranes such as primary cilia and mechano-sensitive ion channels, odontoblasts detect movements of dentinal tubular fluid caused by external stimuli through their processes, transmitting sensory signals from enamel or root surfaces to the nerves in the pulp [33,39,40,41]. In tooth defects, dentin serves as a protective barrier for the dentin-pulp complex, defending against the invasion of external stimuli [11]. Odontoblasts sense abnormalities in fluid shear stress and chemical balance, transmit signals to the underlying neurons, and subsequently initiate innate immune responses to protect the dental pulp from injury [41,42,43]. This anatomical arrangement is fundamental to establishing a unique sensory mechanism with responsive/defensive functions for the tooth. The distinct structural attributes of dentin, encompassing its nano-fibrous/mineralized extracellular matrix (ECM), micro-sized dentinal tubules, and odontoblast polarization and processes, collectively underpin its essential roles in providing physical support and protection and serving as a mechano-sensor in the tooth.

### 2.2. Dentin Mechano-Sensing through Mechano-Sensitive Ion Channels

Mechano-sensation is the process by which cells detect mechanical forces in their environment. Mechano-transduction is the conversion of these mechanical stimuli into biochemical signals within the cells. When cells detect mechanical forces through their mechano-sensory machinery, such as mechano-sensitive ion channels, these forces are translated into a series of intracellular events that can result in changes to cell function, gene expression, or other cellular responses [44,45]. The precise mechano-sensory mechanism underlying the detection of mechanical stimuli by odontoblasts remains incompletely elucidated; however, several theoretical frameworks have been advanced to rationalize the mechano-sensing of odontoblasts. Among various types of receptors in the odontoblast membrane [41], mechano-sensitive ion channels (MICs), including PIEZO and TRP channels within the plasma membrane of odontoblasts, are known to be responsible for mechano-sensing [46] (Table 1). Upon exposure to mechanical stimuli, plasma membranes deform, and the altered membrane tension gates those MICs, subsequently generating intracellular calcium ion influx and initiating mechano-signal transduction [47,48,49,50,51,52,53,54] (Figure 1C). Various downstream cascade signaling pathways are activated, eventually promoting cells to respond to the mechanical stimuli accordingly [49,55,56,57,58,59] (Figure 1D).

#### 2.2.1. Mechano-Sensitive PIEZO Channels

PIEZO channels are a trimeric structure located in the plasma membrane. When mechanical force is applied to the cell membrane, it triggers the opening of the PIEZO channels. This, in turn, allows the influx of extracellular Ca^2+^, facilitating the conversion of mechanical signals into electrical and chemical signals within the cell. PIEZO channels are pivotal in mechano-sensation within tissues and organs reliant on mechanical stimulation for maintaining their function, such as blood vessels, urinary bladder, urethra, skin, and heart [60]. PIEZO1 and 2 channels are recognized as primary sensors for mechanical forces [56,61]. Specifically, PIEZO1 channels are integral to detecting various mechanical forces, including compressive stress, tensile stress, and shear stress [62]. Conversely, PIEZO2 channels are predominantly found in sensory neurons and facilitate non-noxious tactile sensations and proprioception [63]. Moreover, PIEZO2 channels govern mechano-sensation encompassing touch, proprioception, and interoception, and contribute to mediating mechanical allodynia (painful sensation) [64,65,66]. Notably, the simultaneous knockdown of PIEZO1 and 2 channels in mice abolished the baroreflex and diminished the overall mechano-sensitivity of neural cells [67].

**Table 1 ijms-25-05642-t001:** Mechano-sensitive ion channels in the dental sensory system.

Mechanical Stimulation	MIC	Downstream Signaling Pathway	Outcome	Ref.
Compressive stress (direct compression)	PIEZO1	NF-kB signaling pathway	PIEZO1 exerts a transduction role in mechanical stress-induced osteoclastogenesis in hPDLCs	[55]
Compressive stress (HP)		ERK1/2 and p38 MAPK signaling pathway	PIEZO1 is responsible for HP and functions as a factor for the cell fate determination of MSCs by regulating *BMP2* expression	[56]
		WNT/β-catenin signaling pathway	PIEZO1 functions as a mechano-transducer that connects HP signals that promote the odontoblast differentiation and ciliogenesis of SHEDs	[57]
Fluid shear stress	PIEZO1, 2	- Calcium signaling pathway - Integrin β1/ERK1 and WNT/β-catenin signaling pathway	Suppress inflammatory and odontogenic genes in OBs	[49,58]
LIPUS		ERK1/2 MAPK signaling pathway	Enhance the proliferation of DPSCs	[59]
Deformation (stretching displacement)	TRPM7	Calcium signaling pathway	Facilitate intracellular Ca^2+^ signaling in the odontoblastic process	[47]
Vertical displacement using a micropipette	PIEZO1		Induce calcium signaling and reduced dentin mineralization in OBs	[48,49]
Temperature shock (heat)	TRPV1, 2, 3, 4		Primary cultured OBs express TRP channels, contributing to mechano-sensitive sensory transmission	[50]
	TRPV1		- cAMP-mediated crosstalk between CB1 and TRPV1 in OBs leads to Ca^2+^ intracellular accumulation - Functional TRPV1-NCX coupling facilitates Ca^2+^ extrusion, drives dentinogenesis, and maintains intracellular calcium homeostasis	[51]
	TRPV1, TRPA1		- These channels exhibit upregulation during the odontogenic differentiation of hDPSCs - Modulate odontogenic differentiation by regulating intercellular Ca^2+^ concentration	[52]
Temperature shock (cold)	PIEZO1	PIEZO1/TRPA1-pannexin-1-P2X_3_ receptor axis pathway	Mediates sensory transduction in dentinal sensitivity between odontoblasts and neurons	[68]
	TRPA1, TRPM8	Calcium signaling pathway	Increased [Ca^2+^]_i_ in response to TRPA1 and TRPM8 activation, majoring physiological importance for pulpal homeostasis	[53]
	TRPC5		TRPC5 is a cold sensor in intact teeth and originates the transduction in odontoblasts	[54]

MIC, mechanical ion channels; hPDLCs, human periodontal ligament cells; HP, hydrostatic pressure; MSC, Menachem stem cells; SHEDs, stem cells from human exfoliated deciduous teeth; OBs, odontoblasts; LIPUS, Low-intensity pulsed ultrasound; DPSCs, dental pulp stem cells; NCX, Na^+^-Ca^2+^ exchangers; TG, trigeminal ganglia neurons; DRG, dorsal root ganglia neurons; hDPFs, human dental pulp fibroblasts.

The tooth is an organ subject to continuous mechanical stress, suggesting the likelihood of unique mechano-adaptation and -sensing mechanisms, particularly within dentin. The primary function of teeth is to masticate food, creating a swallowable food bolus and providing a masticatory system for incising, tearing, and grinding food [69]. During mastication, direct compression involves the application of force directly onto tooth surfaces or dental structures [69]. Dentin exhibits varied deformation behavior under compression, ranging from brittle to deformable states [70]. Recent findings suggest that PIEZO1 channels also act as a mechanical sensor for odontoblasts and dental pulp stem cells (DPSCs), participating in sensing mechanical forces such as hydrodynamic forces, hydrostatic pressure, and tensile/compressive stress, thereby playing essential roles in homeostasis, odontogenic differentiation, and mineralization [49,56,71]. PIEZO channels may contribute to the maintenance of basal intracellular calcium levels, and injury can disrupt intracellular calcium level homeostasis with fluid shear stress overloading, initiating inflammatory and reparative mechanisms in odontoblasts.

Emerging evidence suggests that PIEZO channels play pivotal roles in mechano-sensory and -responsive functions in odontoblasts. For instance, in response to fluid shear stress, odontoblasts elicit a rapid inward current via PIEZO1 channels and release ATP to transmit signals to adjacent odontoblasts [49]. Similarly, direct cellular deformation (i.e., vertical displacement using a micropipette) can induce robust calcium signaling through PIEZO1 channels, thereby regulating dentin mineralization in odontoblasts accordingly [48,49]. PIEZO channels also play a crucial role in sensory signaling transmission between odontoblasts and trigeminal (TG) neurons. In dentin, both PIEZO1 and 2 channels are expressed in both odontoblasts and TG neurons [72]. Following the cold shock, PIEZO1 channels mediate sensory transduction between odontoblasts and neurons via the PIEZO1-TRPA1-pannexin1-P2X_3_ receptor axis pathway [68]. These findings suggest that odontoblasts convert external mechanical stimuli into biological signals through PIEZO channels and produce electrophysiological responses for sensory signal transmission to adjacent cells to facilitate appropriate mechano-responses to mechanical stress. Given the potential role of PIEZO channels in mechano-sensation and pain signaling transmission between dentinal cells, PIEZO channels could be a key therapeutic target for pain management in dentin-associated diseases.

Interestingly, recent findings suggest that PIEZO1 and 2 channels may play distinctive roles in mechano-sensing and -responding within dentin. Although numerous studies have shown that PIEZO1 and 2 channels share the same overall architecture and domain composition, there are differences in cellular force-transmission pathways, indicating that PIEZO1 and 2 channels do not have exactly the same functional mechanism [73]. In vivo studies have demonstrated that while PIEZO1 channels are involved in mediating dental pain in models induced by high-threshold stimuli, PIEZO2 channels act as a low-threshold mechano-receptor that evokes pain in response to weak stimuli [74,75]. Xu et al. reported that PIEZO1 and 2 channels together regulate inflammatory and reparative responses in odontoblasts in response to fluid shear stress overloading upon dentin injury, with PIEZO1 channels being more predominant in mechano-transduction through the integrin β1/Erk1 and Wnt/β-catenin signaling pathways [58]. Under fluid shear stress overloading, the gene expression levels of both *PIEZO1* and *2* channels were increased, while those of inflammatory (i.e., *IL6* and *TLR 4*), odontogenic (i.e., *DSPP* and *DMP1*), and mineralization (i.e., *BMP2*) markers were down-regulated in odontoblasts. This suggests that PIEZO channels could negatively modulate the expression of odontogenic genes to confine the excessive formation of reparative dentin, thereby maintaining enough space for the pulp cavity. Notably, *PIEZO2* expression was significantly increased under low fluid shear stress stimulation compared to *PIEZO1*, indicating that PIEZO2 channels are low-threshold mechano-sensitive ion channels. Moreover, knocking down PIEZO2 alone led to no changes in genes related to odontogenic/reparative functions (i.e., integrin β1, Erk1, Wnt5b, and β-catenin *genes*), unlike PIEZO1/2 double knockdown. These findings suggest that both PIEZO1 and 2 play critical roles in the mechano-sensing and -responding of odontoblasts to fluid shear stress but regulate it through distinctive downstream signaling pathways. Similarly, Matsunaga et al. reported that mechanical stimulation predominantly activates intracellular calcium signaling via PIEZO1 channel opening, rather than PIEZO2 channels, establishing intercellular odontoblast-odontoblast communication [48]. As an odontoblast was directly deformed by aspiration, neighboring odontoblasts showed a transient increase in intracellular calcium level, while cells far away from the mechanically stimulated cells showed a decreased calcium level [48]. PIEZO knockdown completely abolished the mechanical stimulation-induced calcium signaling, suggesting that PIEZO channels may regulate physical crosstalk between odontoblasts in dentin.

In addition to mechano-sensation, emerging evidence suggests that PIEZO channels regulate dentin development, (re)mineralization, and periodontal/orthodontic tooth movement in a mechano-sensitive manner. For example, Matsunaga et al. found that mineralization induced by odontoblasts is negatively correlated with PIEZO1 channel activity, suggesting that PIEZO1 channels suppress physiological and reactional dentinogenesis in response to mechanical stress [48]. In dentin development, PIEZO1 expression levels vary depending on the developmental stage. The expression level of PIEZO1 is low in progenitor cells but significantly increases with odontogenic differentiation in vitro [76]. In vivo studies have demonstrated that PIEZO1 channels are hardly expressed in the tooth germ of mice before birth, but weakly positive expression is detected in the postnatal period, possibly related to the mechanical stimulus of sucking and chewing in the oral cavity of mice after birth [49]. Similarly, numerous in vitro studies have demonstrated that PIEZO1 channels can determine the cell fate of dental progenitor cells in response to mechanical stress. For example, under compressive stress, PIEZO1 channels mediate mechanical stress-induced osteoclastogenesis in human periodontal ligament cells via the NF-kB signaling pathway [55]. PIEZO1 channels are also responsible for sensing hydrostatic pressure generated in dentinal tubule fluid [57]. In response to hydrostatic pressure, PIEZO1 expression was significantly increased and promoted odontoblast differentiation of stem cells from human exfoliated deciduous teeth via the Wnt/β-catenin signaling pathway [57]. Moreover, PIEZO1 channels induce ciliogenesis by inhibiting cell proliferation, suggesting it’s another regulatory role in odontoblast mechano-sensing.

In conclusion, while the exact functions of PIEZO channels and related mechanisms in the dental sensory system remain to be more clearly elucidated, PIEZO channels are indeed predominant mechano-sensitive ion channels, playing crucial roles in dental health through mechano-sensing/-responding.

#### 2.2.2. Mechanical/Thermal-Sensitive TRP Channels

Transient receptor potential (TRP) channels are calcium-selective cation channels activated by mechanical or temperature changes and expressed in various dentinal cells, including odontoblasts, dental pulp fibroblasts, and dental pulp stem/progenitor cells [51,53,54,77,78,79]. While PIEZO channels predominantly respond to mechanical stimuli, multiple other mechano-sensitive ion channels, such as TRP channels, are simultaneously involved in mechano-sensing and signal transduction [46]. For instance, upon the activation of TRP channels (TRPV1, TPRV2, TRPV4, and TRPA1) by mechanical stimulation, odontoblasts release ATP as a neurotransmitter through the ATP-permeable channel, pannexin-1. This signal is received by adjacent TG neurons in pulpal nerve endings through P2X_3_ receptors [80]. Moreover, TRP channels are functionally associated with PIEZO channels and other sensory receptors in mechanical stimulation-mediated dentinogenesis. PIEZO1-mediated calcium signaling, activated by fluid shear stress, leads to the opening of TRPV4 channels, sustaining calcium signaling [81]. Additionally, functional coupling between TRPV1 channels, cannabinoid (CB) receptors, and Na^+^-Ca^2+^ exchangers (NCXs) enhances nociception in odontoblasts [51]. Upon activation by external stimuli, cAMP-mediated crosstalk between CB1 and TRPV1 facilitates intracellular calcium accumulation. To prevent excessive calcium levels, functional coupling between TRPV1 channels and NCXs aids in extruding surplus calcium, thereby maintaining calcium homeostasis and driving dentinogenesis [51].

Notably, distinct physical distributions of TRP and PIEZO channels within the cellular cytoplasm of odontoblasts have been observed. While PIEZO1 and 2 channels are expressed throughout odontoblasts, including the cell body and processes, TRPM5 channels are asymmetrically distributed, with distinct localization to regions proximal to and within odontoblast processes [49,72]. Additionally, TRPA1 channels are expressed in the terminal parts of odontoblasts and within odontoblast processes [82]. This observation suggests that these cellular regions may represent specialized sensory compartments [83]. Upon mechanical stimuli, activation of mechano-sensitive calcium channels increases intracellular calcium concentrations, and subsequent Ca^2+^-dependent opening of TRPM5 channels could induce depolarization within odontoblasts [83,84], potentially contributing to the functional behaviors of odontoblasts. These findings have significant implications for understanding how cells efficiently sense and crosstalk within local odontoblast networks and synaptic networks associated with odontoblasts.

Mounting evidence indicates that mechano-sensitive responses from odontoblasts through mechano-sensitive ion channels and subsequent nociceptive transduction in pulpal nerves contribute to the pain associated with dentinal diseases [21]. Furthermore, mechano-sensitive ion channels not only function as receptors but could also direct the cell fate of dentinal stem/progenitor cells [51,85]. Future research should prioritize uncovering the precise mechanisms through which mechano-sensitive ion channels regulate the dental sensory system. This understanding will pave the way for the development of effective treatments for managing pain in diseases associated with dentin.

## 3. Advanced Dentin-Mimicking In Vitro Models

In dentin mechano-sensing, odontoblasts serve as sensory transducers, detecting external stimuli and transmitting signals to the nerves, which is fundamental to the sensory mechanism underlying tooth pain. Dentin possesses a unique architecture for mechano-sensing, as described in Section 2.1. Briefly, the dentinal ECM has a hierarchical structure, with micro-sized tubules/canaliculi embedded in a highly interconnected nanofibrous three-dimensional (3D) network [86]. More importantly, odontoblasts display a unique cell morphology, characterized by a columnar cell body aligned at the periphery of the dental pulp and a long cytoplasmic process extending from the cell body to a dentinal tubule. This unique architecture enables odontoblasts to efficiently sense mechanical stimuli in the dentin sensory system.

In vivo models are not suitable for dissecting real-time events in mechano-sensing and -responding, and the complexity of interpreting phenomena observed in vivo limits our understanding. Therefore, various in vitro model systems for dentin, including tissue culture plates, the Hume model, in vitro pulp chambers, and dentin barrier tests, are extensively used. However, these models have limited ability to closely mimic the physical and mechanical features of dentin tissues. For example, odontoblasts lose their long processes when cultured as a monolayer on a tissue culture plate [87,88]. Moreover, in dentinal tubules, small forces (~90 Pa) can open mechano-sensitive TRP channels in odontoblasts, while a much larger force (~2800 Pa) is required to open the same channels in the in vitro monolayer model [54]. These findings indicate that odontoblast morphology, especially processes, is a crucial factor in determining mechano-sensitivity and mechano-transduction. Additionally, they emphasize the necessity of developing in vitro platforms with dentin-specific structures that can induce the native morphology of odontoblasts. This would improve our comprehensive understanding of dentin-related diseases from a mechanobiological standpoint, thus aiding in the development of novel treatment strategies.

In this section, we explore the current state of dentin-mimicking in vitro models, including microchannel platforms replicating dentinal tubule structure, fibrous platforms mimicking the hierarchical architecture of the dentin matrix, and tooth-on-a-chip models mimicking the spatial arraignment of dentin-pulp (see Table 2 and Figure 2). We emphasize their utility in uncovering mechanobiological insights and their importance in the field of dentistry.

**Table 2 ijms-25-05642-t002:** Advanced dentin-mimicking platforms.

Platform	Design		Goal	Findings	Ref.
**To mimic** **the microchannel structure of the** **dentinal tube**					
Microfluidic chip	Microchannel size was 3 µm *H*, 150 µm *L*, and different *W* (2, 4, 6, and 8 µm)		To investigate the relationship between the growth of odontoblast processes and the geometric constraint imposed by microchannels	2 μm is the appropriate size for inducing the growth of odontoblast processes in vitro	[89]
Compression bioreactor for cyclic loading of the microhole constructs	Silicon membranes with pores, 10 μm depth × 5 µm diameter × 10 μm pitch; cell density, 4.0 × 10^5^ cells/cm^2^; compression magnitude, 19.6 kPa, at a frequency of 0.083 Hz for 9 h followed by seeding to the membrane for 16 h		To study the mechanical and geometrical cues on odontoblast differentiation and processing of stem cells	Odontoblastic differentiation of hDPSCs is promoted by optimal mechanical compression through the MAPK signaling pathway and the expression of the *BMP7* and *WNT10A* Without micropores or mechanical compression, odontoblast differentiation and processing barely occur	[90]
**To mimic the** **hierarchical architecture of the** **dentin matrix**					
Fibrous structure matrix	Electrospinning of PSF matrix was of meshes with diameters of 300~500 nm		To investigate the effect of nanofibrous topology on DPSC differentiation to odontoblasts	The nanofibrous topology contributes to the odontogenic differentiation of hDPSCs by enhancing Wnt/β-catenin signaling	[91]
				Also contributes to osteoblast differentiation by stabilizing the Runx2 protein compared to those in the non-fibrous matrix	[92]
	w/micro patterns	Gelatin nanofibrous matrix with microholes at a density of 20,000/mm^2^ and a pore size of 2–5 µm	To mimic highly ordered tubules with similar size and density as natural dentin	The DPSCs were highly polarized on the tubular matrix along with up-regulated odontoblast differentiation markers (*ALP*, *COL1A1*, and *DSPP*) compared to cells cultured on a non-tubular matrix	[86]
		Gelatin electrospun nanofibers with a diameter of 200 nm → Lithographing with circular microislands with a diameter of 60 μm → Laser ablation creating a microhole with a diameter of 3–4.5 µm in the microisland	To develop a physiologically relevant 3D platform to recapitulate the morphologies of odontoblasts in vitro	Established a unique platform that mimics hierarchic 3D nanofibrous tubular/canaliculi architecture by combining electrospinning, photolithography, and laser ablation techniques	[93]
			To understand the underlying mechanisms of DPSC polarization by decoupling various physical factors that control cell polarization	Both actin and microtubules were critical to DPSC adhesion and polarization RhoA/ROCK signaling pathway is involved in regulating DPSC polarization	[94]
Microfluidic chip-based system	The microchannel size was 5 µm *H* and 10 µm *W*		Allow for the application of specific media for the appropriate development of neuronal and dental tissues	Neurites are repealed when co-cultured with embryonic tooth germs, while postnatal teeth exert an attractive effect on trigeminal ganglia-derived neurons	[95]
Tooth-on-a -chip	The device consists of four reservoirs with an 8 mm punch and two parallel channels, two perfusable chambers (300 μm *W* × 1 mm *L* × 1 mm *H*), and a central groove that holds a dentin fragment		To replicate physiological blood flow dynamics and the intricate interface between dental pulp and dentin	- SCAPs presented consistently higher metabolic activity on-chip than off-chip - A real-time tracking system capable of culturing cells and investigating gelatinolytic activity while simultaneously imaging the HL as formed by the adhesive system	[96]
	The device has a central cylindrical chamber to house the dentin disc. Beneath the central chamber is a rhomboid-shaped perfusable microchannel with circular openings at either end		To characterize the cytotoxicity potential of SDF on DPSCs and gingival equivalents	The gingiva-on-chip presented models of the microphysiological features of the gingival tissue in healthy and diseased states, and their interface with the external milieu	[97,98]

H, height; L, length; W, width; hDPSCs, human dental pulp stem cells; PSF, polystyrene nanofibrous; ECM, extracellular matrix; DPSCs, dental pulp stem cells; HAP, hydroxyapatite; HA, Hyaluronic acid; HA-MCS, hyaluronic acid-modified collagen scaffolds; DRGs: dorsal root ganglion; aDESCs; adult dental epithelial stem cell; DEO, dental epithelial organoids; pDE, porcine dental epithelial cells; HMPs, hydrogel microparticles; DE-DM, dental epithelium-dental mesenchyme; SCAPs, stem cells from the apical papilla; SDF, silver diamine fluoride.

**Figure 2 ijms-25-05642-f002:**
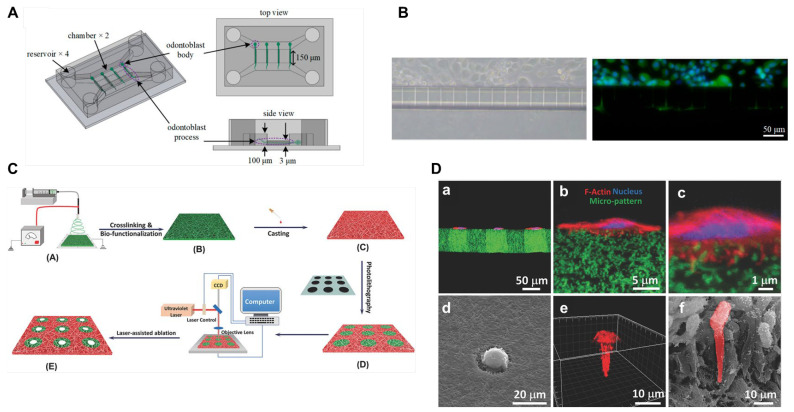
Advanced dentin-mimicking in vitro models. (**A**) The design of the microfluidic chip, comprising four reservoirs, two chambers, and hundreds of microchannels connecting them. (**B**) The morphologies of odontoblasts within the chip. An optical microscope image (**left**) and an immunofluorescence image (**right**) demonstrate that odontoblasts extend their processes along the microchannels (green: Aquaporin 4, blue: nucleus). (**C**) The fabrication process of a nanofibrous tubular 3D platform that integrates micropatterned microstructures into biomimetic 3D fibrous scaffolds. This innovative approach combines nanofabrication, micropatterning, and computer-assisted laser ablation to create a biomimetic nanofibrous tubular 3D matrix. (**D**) A cross-section view of DPSCs cultured on the matrix. (**a**–**c**) fluorescence images presenting cross-section views (red: F-actin, blue: nucleus, green: micropattern). (**d**) scanning electron microscope (SEM) images and (**e**) fluorescence images of a DPSC cultured on a microisland, showcasing highly polarized morphologies on the tubular matrix. (**f**) SEM images of the morphology of odontoblast in vivo (pseudo-colored in red). References [89,93] provide further details, with permission from ACS Publications and Wiley Publishing Group, respectively.

### 3.1. Microchannel Platforms Mimicking Dentinal Tubule Structure

Odontoblasts possess a unique morphological feature known as the process. These processes extend from the odontoblast body located at the periphery of the pulp into the dentin tubule [99,100], enabling odontoblasts to sense external stimuli effectively [101,102]. External stimuli can cause the deformation of dentin tubules and the flow of dentinal fluid, generating fluid shear stress that stimulates odontoblast processes. Subsequently, odontoblast processes release neurotransmitters to adjacent nerves, leading to sensation and even pain [80]. Therefore, in vitro models of odontoblasts should accurately replicate this process structure to enhance our understanding of dental physiology and pathology.

However, modeling the growth of odontoblast processes in vitro presents challenges. Despite various culture systems developed to date, in vitro odontoblast morphology still differs from that in vivo. Most studies have been conducted using traditional tissue culture plates, where odontoblasts lack processes [103]. Recent efforts have focused on developing platforms capable of inducing odontoblast processes-like structures by introducing micropatterns [89,90]. Micropatterns such as microchannels or microholes mimic the geometric constraints of the dentin tubules and promote the growth of odontoblast processes. For instance, microfluidic chips fabricated via soft lithography with microchannels closely mimic the microstructures of dentin tubules, successfully inducing the growth of odontoblast processes from cell bodies [89] (Figure 2). Importantly, the geometric constraints imposed by microchannels play a pivotal role in the growth of odontoblast processes. The growth of odontoblast processes was successfully induced by microchannels with a diameter of 2 µm, whereas cells migrating through the channels with widths of 4 µm, 6 µm, and 8 µm did not exhibit process growth. This result suggests that a channel diameter of 2 µm is optimal for inducing the growth of odontoblast processes in vitro. Moreover, another study demonstrated that the diameter of microtubules significantly influences microbial colonization. When the diameter is greater than 2 µm, bacterial migration and colonization are largely accelerated [104].

In another study, a custom-made scaffold composed of a silicone membrane, designed to mimic dentinal tubules, featured micropores with dimensions of 10 µm depth × 5 µm diameter × 10 µm pitch. This scaffold was utilized to investigate the effects of mechanical and geometrical cues on the differentiation and processing of human DPSCs in vitro [90]. The evaluation was confirmed by the mRNA expression of the odontoblastic markers *DSPP* and *MMP20* (enamelysin), and the presence of typical morphological features of odontoblasts, including alignment in a parallel configuration and extension of cell processes into the pores. To further explore the impacts of mechanical compression on odontoblast differentiation, hDPSCs cultured on the silicon membranes were subjected to cyclic compression loading using a compression bioreactor. Under optimal conditions (cell density of 4.0 × 10^5^ cells/cm^2^; compression magnitude of 19.6 kPa at a frequency of 0.083 Hz for 9 h followed by seeding onto the membranes for 16 h), there was a significant increase in the expression of odontoblast-specific markers, DSPP, and enamelysin, along with enhanced elongation of cellular processes into the pores of the membrane, indicative of odontoblast differentiation. Additionally, mechanical compression led to up-regulation of *BMP7* and *WNT10A*, as well as enhanced phosphorylation of MAPKs, ERK1/2, and p38, indicating that optimal mechanical compression promotes odontoblast differentiation of hDPSCs through the MAPK signaling pathway.

Intriguingly, human mesenchymal stem cells derived from bone marrow (hBMSCs) and amnion (hAMSCs) also differentiated into odontoblasts in response to optimal mechanical compression, underscoring the importance of the physical structure of the scaffold in directing stem cell lineage and fate towards odontoblasts. The application of mechanical compression on the biomimetic silicone membrane unexpectedly promoted odontoblast differentiation in hBMSCs and hAMSCs, similar to the observations in hDPSCs. Conversely, cells showed minimal differentiation without mechanical compression on the membrane with pores or with mechanical compression on the membrane lacking pores. These results demonstrate that silicon membrane substrates with dentinal tubule-like geometry exert a significant influence on stem cell lineage and fate determination. It is plausible that scaffold geometry, along with mechanical compression, could modulate cell surface receptors, thereby initiating the activation of the MAPK signaling pathway [105,106,107]. Such intricate interplay between scaffold properties and mechanical cues holds immense potential for influencing cellular behavior and directing tissue regeneration strategies in dentistry and beyond.

In terms of methodology, microchannels can be fabricated using the standard soft lithography technique, a commonly employed method for microfluidic chip fabrication. This technique offers excellent repeatability and enables high-throughput fabrication. Microchannels can be designed with various lengths (e.g., >150 µm) and shapes (e.g., linear vs. trapezoidal) to induce odontoblast-specific morphology [108]. Microchannel chips typically induce process growth in a horizontal direction, while microholes tend to generate vertical growth. However, whether the direction of process growth on in vitro platforms contributes to the functioning of odontoblasts warrants further investigation.

While various micropatterned platforms have successfully mimicked dentin tubules, the growth of odontoblast processes on these platforms relies on defined geometric constraints. However, in vivo, dentin tubules develop concurrently with odontoblast differentiation from their progenitors [109]. During this process, odontoblast progenitors simultaneously form a tubular structure and extend their processes. Therefore, current platforms may only replicate the structure and morphology of dentin tubules. To address this limitation, future research efforts should prioritize the development of platforms that stimulate the spontaneous formation of dentin tubules by odontoblasts rather than relying on pre-formed guidance for odontoblast processing. Additionally, future strategies could explore the integration of microchannels with advanced tissue engineering technologies, including organoids, micro-mimetics, and 3D bioprinting [110,111].

Furthermore, although it has been demonstrated that odontoblasts with process-like structures induced by micropatterns promote odontoblast differentiation of dentinal stem cells and increase odontoblast-specific gene expression, the potential improvement in cellular functionality, particularly mechano-sensing ability, has not been investigated thoroughly. Therefore, future research should concentrate on assessing the effects of odontoblast process formation on mechano-sensing and mechano-transduction for a more comprehensive understanding.

Nonetheless, these dentinal tubule-mimicking platforms hold significant promise as powerful tools for modeling dental diseases such as dentin hypersensitivity, which is related to odontoblast processes. Furthermore, they offer opportunities to investigate the physiology and pathology of odontoblast processes, thereby contributing to the development of treatment solutions for dental diseases.

### 3.2. Fibrous Platforms Mimicking the Hierarchical Architecture of Dentin Matrix

Let us review the structure of dentin (Figure 1A). Odontoblasts in dentin reside within a distinct 3D microenvironment—a highly structured ECM characterized by a nanofibrous network with a well-organized hierarchical architecture ranging from nano to microscales. However, when odontoblasts are cultured on traditional two-dimensional (2D) tissue culture plates, they rapidly lose their characteristic dendritic shapes, indicating the loss of physiological morphologies for odontoblasts [112]. This highlights the limitation of traditional cell culture methods in replicating the native microenvironment of dentin, which is crucial for maintaining the proper function and behavior of odontoblasts.

Recently, platforms based on synthetic nanofibrous scaffolds have been developed to investigate the effect of dentin matrix architecture, especially fibrous topology and tubular structure, on odontoblast differentiation, processing, and functions [113]. For instance, Rahman et al. investigated the impact of fibrous topography on directing the odontoblast differentiation of DPSCs [91]. Considering that tissue culture plates are made of polystyrene (PS), a PS fibrous matrix was prepared by electrospinning to provide a fibrous topology with precise control over fiber density, diameters, and alignment. When DPSCs were cultured on the fibrous matrix, the expression of odontoblast-specific markers, such as *DSPP*, *BGLAP* (osteocalcin), and *IBSP* (bone sialoprotein), was significantly higher compared to tissue culture plates with flat surfaces. Furthermore, cells cultured on the PS fibrous matrix promoted the expression of *WNT* (*3A*, *5A*, and *10A*), *BMP* (*2*, *4*, and *7*), and *CTNNB1* (β-catenin) and its transcriptional activity. However, when the WNT3A-initiated signaling was blocked, the fibrous matrix-induced DSPP expression was abrogated, suggesting that the fibrous topology strongly supports odontoblastic differentiation of DPSCs by enhancing WNT/β-catenin signaling. Moreover, the fibrous structure of the matrix contributes to odontoblast differentiation by stabilizing the Runx2 protein, thus decreasing the ubiquitin-dependent degradation of Runx2 in pre-osteoblast compared to those in tissue culture plates or a non-fibrous collagen matrix [92]. This result again highlights the importance of the fibrous topology of the dentin matrix in improving odontoblast differentiation.

Considering the complexity of in vivo environments, in vitro platforms that allow the decoupling of factors controlling DPSC polarization can greatly aid in understanding underlying mechanisms and designing new approaches for dental research and regeneration therapy. Odontoblast polarization is crucial for odontoblast differentiation and the formation of tubular dentin, which is essential for maintaining normal tooth function and regenerating dentin. During regenerative endodontics, DPSCs must undergo polarization and retain their polarized status to function as odontoblasts. Since DPSC polarization is regulated by a dynamic signaling network comprising biophysical and biochemical factors as well as cell-cell interactions, in vitro platforms that induce 3D physiological morphologies of dentinal cells by mimicking the biophysical features of dentin are vital for studying dentin mechanobiology.

Recently, micropatterned synthetic matrices precisely mimicking the hierarchic 3D nanofibrous tubular/canaliculi architecture have been developed as a physiologically relevant 3D platform [93]. The Liu group has pioneered a unique nanofibrous tubular 3D platform integrating micropatterned microstructures into biomimetic 3D fibrous scaffolds as dentin-mimicking platforms [86,93,94] (Figure 2C). They established an innovative approach integrating nanofabrication, micropatterning, and computer-assisted laser ablation to create a biomimetic nanofibrous tubular 3D matrix. For instance, a micropatterning method employing laser guidance, a non-contact, high-precision, flexible computer-programmed machining process, can create highly ordered tubules with densities ranging from 1000–60,000/mm^2^ and sizes varying from 300 nm to 30 µm in a 3D nanofibrous matrix [86]. The synthetic tubular gelatin matrix had a tubule density of 20,000/mm^2^ and pore sizes of 2–5 µm, mimicking the human dentin matrix for tubular dentin regeneration. DPSCs exhibited highly polarized morphologies on the tubular matrix, with almost all tubules of the gelatin matrix occupied by processes of DPSCs after a long-time culture (7 days) (Figure 2D). In contrast, DPSCs cultured on the non-tubular matrix presented branched morphologies with several randomly oriented protrusions attached to nanofibers on the matrix surface without major processes forming [86]. Additionally, odontoblastic differentiation makers (i.e., *ALP*, *COL1A1*, and *DSPP*) were enhanced by culture in the tubular matrix. This tubular architecture provides pivotal biophysical cues controlling the alignment, migration, polarization, and differentiation of DPSCs. Importantly, when implanted in vivo, the 3D tubular hierarchical matrix successfully regenerated functional tubular dentin with well-organized microstructures resembling its natural counterpart.

In another study, a similar platform was used to explore the polarization of DPSCs at a single-cell level. To eliminate the interference of cell-cell contact communications, a 3D platform confining a single dental pulp stem cell was fabricated by combining electrospinning and photolithography techniques. It began with the fabrication of a nanofibrous matrix made of methacrylate-modified gelatin (GelMA) to generate a dentin ECM-like nanofibrous structure. The chemical composition of gelatin is similar to that of collagens abundant in the natural dentin ECM [114,115]. After electrospinning, polyethylene glycol diacrylate was cast onto the GelMA matrix surface, rendering it non-adhesive for cells. Subsequently, cell adhesive areas, referred to as microislands, were selectively created by blocking areas from UV irradiation using a photomask with circular microisland patterns with a diameter of 60 µm. Finally, a microhole with a diameter of 3 µm was created in the center of each microisland using computer-aided laser ablation. Thus, this platform allowed single-cell adhesion to the microislands, where gelatin fibers provided cell adhesion sites and ECM-like fibrous structural cues. Furthermore, the 3 µm-sized microhole, similar to a natural dentinal tubule, allowed cell polarization in 3D. This platform is versatile; for example, the shape and size of microislands can be precisely controlled by photomasks, and the tubular size, distribution, and density can be accurately matched to those of natural dentin ECMs by laser-guided ablation, a non-contact, high-precision, and computer programmed machining process. Additionally, the tubular structure, including width and depth, can be controlled by laser power, laser writing speed, and pulse frequency. This suggests that this approach enables the mimicking of both physiological and pathophysiological morphologies of dentinal cells for mechanobiology studies.

By confining a single DPSC in each microisland, researchers could observe the dynamic sequential process of DPSC polarization and identify the crucial role of nanofibrous tubular architecture in initiating polarization at a single-cell level [94]. They found that the nanofibrous tubular architecture was crucial for initiating the polarization of DPSCs. Dynamic morphological observations showed that the cellular process of polarized DPSCs continuously extended and reached a plateau at 72 h. Meanwhile, the Golgi apparatus, a cell polarization marker, continuously moved from a juxtanuclear region, passed the nucleus, and eventually settled down at a final position a few micrometers away from the nucleus. Disruption of actin and microtubule polymerization resulted in the failure of the adhesion and polarization of DPSCs, although the effect of microtubules was less prominent than actin polymerization. The RhoA/ROCK signaling pathway, which regulates the organization and distribution of cytoskeletal elements and modulates intracellular tension, is particularly involved in modulating polarization, not adhesion. Together, this study demonstrated the indispensable role of cytoskeleton reorganization in modulating the polarization of DPSCs. These findings expand the understanding of DPSC polarization and contribute to the design of new bioinspired materials for regenerative endodontics.

In conclusion, these dentin matrix-mimicking platforms effectively support cells structurally and provide them with a suitable microenvironment for dentin regeneration, serving as valuable tools for studying the physiology and pathology of dentinal cell behaviors and developing treatments for dental diseases. By utilizing these designed in vitro platforms, researchers can gain valuable insights into the influence of intrinsic mechanics and external mechanical stimuli on the regulation of dentinal cell functional behaviors, offering a pathway for investigating the intricate interplay between matrix mechanics and cellular responses in dental research.

### 3.3. Tooth-on-a-Chip: Platforms Mimicking Spatial Arrangement of Dentin-Pulp Complex

Various stimuli, such as thermal, mechanical, and chemical factors, applied to the exposed surface of dentin induce pain. The primary mechanism underlying dentinal sensitivity has been elucidated through the “hydrodynamic theory” [116,117], which posits that stimuli on the dentin surface induce fluid flow into the dentinal tubules. This flow directly and mechanically stimulates not only the nerve endings but also the odontoblasts, the sensory receptor cells located at the dental pulp end of the tubules. Recent in vitro evidence, as discussed in Section 2, demonstrates the mechano-sensing mechanisms in odontoblasts and the neurotransmission mechanisms between odontoblasts and neurons that modulate sensory signal transmission for dentinal sensitivity and pain [80,118,119,120,121,122]. Thus far, in vitro findings have been obtained using the traditional co-culture method on tissue culture plates, and mechanical deformation is applied to odontoblasts using patch clamp techniques, while electrophysiological responses in nearby neurons are selectively recorded. While this approach provides precise control over deformation and allows the recording of relevant neurotransmission, its yield is very low [123]. Moreover, this in vitro method fails to provide spatial compartmentalization between odontoblasts and neurons as observed in vivo in the dentin and pulp.

Recently, to understand the sequential signal transmission in dentin, in vitro platforms mimicking the spatial arrangement of dentin and pulp have been developed based on compartmentalized microfluidic devices [95,124,125]. For example, Franca et al. developed a microfluidic model of the intricate interface between dentin and pulp and used it to determine the real-time response of pulp cells to various dental materials such as phosphoric acid, adhesive monomers, and dental adhesive systems [96]. One device consists of four reservoirs with an 8 mm punch and two parallel channels, accompanied by two perfusable chambers measuring 300 µm in width, 1 mm in length, and 1 mm in height. Additionally, it features a central groove designed to secure a dentin fragment, facilitating the cultivation of dental cells under dynamic flow conditions. Cells displayed consistently higher metabolic activity on-chip than off-chip. Moreover, the chip provides direct visualization of the complexity of the dentin-pulp-biomaterials interface and enables real-time assessment of the response of pulp cells to dental materials on a level previously not possible.

In another study, a microfluidic device was constructed through the thermal bonding of four microstructured poly(methyl methacrylate) (PMMA) sheets in a vacuum oven. Briefly, the device comprises a centrally positioned cylindrical chamber, referred to as the dentin chamber, measuring 12 mm in diameter and 1 mm in length, designed to accommodate a dentin disc securely clamped by silicone O-rings. Adjacent to the dentin chamber is a cylindrical hollow body featuring a removable lid with a diameter of 4 mm, serving as a reservoir for loading test substances. Situated below the central chamber is a rhomboid-shaped perfusable microchannel, termed the pulp channel, measuring 38.5 mm in length, 1 mm in height, and possessing a variable width of either 7 mm or 1 mm at its maximum or minimum dimensions, respectively. Circular apertures at both ends of the pulp channel facilitate cell seeding and the perfusion of reagents [98]. By using this custom-designed microfluidic device fitted with dentin discs of varying thickness (0.5, 1.0, and 1.5 mm) and peristaltic media flow conditions, researchers aimed to characterize the cytotoxicity potential of silver diamine fluoride (SDF) on DPSCs and gingival equivalents [98]. The investigation revealed that SDF exhibited cytotoxic effects on cells at notably low concentrations (0.001%) and was demonstrated to penetrate dentin of low thickness (≤1.0 mm), thereby inducing adverse effects on cells in vitro. Additionally, SDF was observed to disrupt gingival epithelial integrity, resulting in mucosal corrosion [98]. Likewise, the platform closely replicates clinical situations, and thus many recent studies have used such compartmentalized microfluidic devices for dental research. However, most of them primarily use the platforms to investigate the cytotoxic effects of dental materials [126,127,128,129].

Indeed, the tooth-on-a-chip platform shows promise in replicating the near-physiological conditions of the dentin-pulp interface in vivo and enables precise control over biophysical properties such as spatial compartmentalization and microchannel shape and size. Furthermore, such platforms allow real-time cellular responses to dynamic mechanical stimuli, which were limited in existing model systems in dental research [130,131,132,133,134,135,136,137]. Unfortunately, the field of tooth-on-a-chip has remained largely underdeveloped. Future studies should focus on developing in vitro platforms that mimic the intricate physical complexity of natural dentin. Based on that, we can further expand our understanding of dental mechanobiology and therapeutic strategies.

## 4. Concluding Remarks and Future Perspective

While the research topic of “dentin mechanobiology” has garnered increasing attention recently, our understanding is constrained by a scarcity of research articles. Existing materials and platforms primarily focus on mimicking chemical properties rather than the physical features of dentin. Although our discussion, here, primarily addresses potential rather than actual mechanobiological findings, we believe that, considering its significance and importance, dentin mechanobiology will emerge as a crucial research field in dentistry in the near future.

Although there are many theories about how exactly dentin senses external stimuli, the hydrodynamic theory is the most widely accepted. However, this theory alone does not explain all aspects of the process. Fluid mechanical modeling of dentinal microtubules provides mechanistic insights into dental pain [138,139,140]. Unfortunately, most current models primarily focus on thermal sensing rather than other mechanical stimuli [102,141,142,143]. Future research needs to address this gap by exploring how dentin directly senses mechanical forces.

Advancements in in vitro platforms that faithfully mimic dentin structure have provided deeper insights into the mechano-sensitive nature of odontoblasts within dentin and their responses to mechanical stimuli under both physiological and pathological conditions, such as dentin-associated diseases. While traditional monolayer culture systems have been the cornerstone of dental research, the field has now expanded to encompass advanced in vitro platforms that closely replicate physical features found in vivo, including three-dimensionality, nano-/microscale topologies, hierarchical structures, and mechanical dynamics. Increasing evidence suggests that dental cells cultured in dentin-mimicking in vitro platforms exhibit functional characteristics more akin to their in vivo counterparts compared to those cultured in monolayer systems. This paradigm shift underscores the significance of replicating the physical features of dentin in vitro as a crucial step toward better understanding pathogenesis and establishing effective therapeutic strategies for dentin-associated diseases.

Nevertheless, significant challenges persist, especially in replicating the complex chemo-physical properties of dentinal ECM within in vitro setups. Current platforms struggle to integrate a myriad of structural, mechanical, and chemical cues of the matrix within a single setup. Some studies have attempted to mimic the chemo-physical features of dentin by incorporating mineral components into fibrous matrices, demonstrating enhanced proliferation and differentiation of DPSCs in vitro, leading to in vivo dentin regeneration [144]. However, the combined and synergistic effects of physical and chemical cues on dentinogenesis and pathogenesis have not been extensively investigated.

Furthermore, the mechanical properties of matrices, such as stiffness, stress-relaxation, and plasticity, have not been adequately considered in dentin-mimicking in vitro platforms. Emerging evidence suggests that cytoskeletal remodeling of odontoblasts influences mechano-transduction, thereby impacting the odontoblast phenotype [85,145]. For example, a fibrous matrix with lower stiffness (kPa) supports greater odontoblast differentiation of dentinal stem cells compared to traditional culture substrates (TCPs) with much higher stiffness (GPa) [146]. Similarly, biomineralized collagen fibrous matrices with a stiffness similar to that of natural dentin enhance the proliferation and differentiation of DPSCs into odontoblasts compared to their non-mineralized counterparts [144]. These recent findings strongly imply that the mechanical properties of matrices could govern the mechano-sensitivity and mechano-responses in dentin, similar to observations in other mechano-sensitive tissues/organs [147]. Furthermore, in restorative dentistry, material bonding to dentin presents greater challenges compared to bonding to enamel due to dentin’s complex structure and mechanics [148]. At the adhesive-dentin interface, the local micro/nano-structures and -mechanics play a crucial role in determining both initial bonding effectiveness and long-term bond stability [149,150]. This again underscores the significance of comprehending the physical and mechanical properties of dentin and incorporating them into the material design for successful dental regenerative therapy.

Moreover, the intricate compartmentalization of the dentin matrix with different cell types presents a challenge in developing dentin-mimicking in vitro platforms. While micropatterned platforms can mimic the architectural arrangements of dentin, dentinal cells are still confined to 2D physical contacts. Tooth organoids offer an alternative for replicating 3D spatial-temporal cell-cell interactions [151,152,153]. However, organoid development relies on self-assembly, resulting in random cell mixing and positioning within organoids without spatial arrangement between cells. This challenge could potentially be addressed by emerging tooth-on-a-chip technologies [153].

Unlike bone, which has made significant advancements in mechanobiology research [154,155,156,157], dentin research is still in its nascent stages. However, given the similarities between odontoblasts in dentin and osteocytes in bone, particularly in their mechanobiological behaviors, insights from bone research could offer potential solutions to overcome current challenges in developing dentin-mimicking platforms for mechanobiology studies.

In conclusion, advanced dentin-mimicking in vitro platforms are crucial for elucidating fundamental mechanobiological phenomena and, more importantly, for paving the way toward the development of effective mechano-therapeutic strategies for dentin-associated diseases. Continued efforts in this field hold great promise for advancing our understanding of dentin mechanobiology and improving clinical outcomes for patients with dentin-associated conditions. 

## Figures and Tables

**Figure 1 ijms-25-05642-f001:**
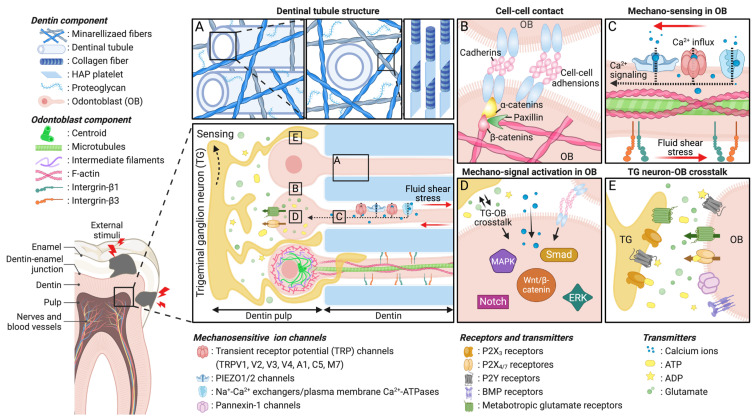
Schematic of mechano-sensing/transduction/transmission in dentin. Odontoblasts adopt a cylindrical shape and exhibit structural polarity, forming the outermost cell layer of the dental pulp tissue, which is advantageous for their role as sensory transducers. The nerve endings of dental primary afferents extend into dentinal tubules, establishing a distinct sensory mechanism for the tooth. (**A**) Diagram depicting the hierarchical structure of dentin’s extracellular matrix, including dentinal tubules, and mineralized nanofibrous networks. (**B**) Odontoblast bodies align at the periphery of the dental pulp and establish physical contact with nearby odontoblasts, forming an intercellular odontoblast-odontoblast network. (**C**) Odontoblasts sense mechanical stimuli through mechano-sensitive ion channels, such as PIEZO and TRP channels, within their processes located in dentinal tubules. (**D**) Upon mechano-sensing, mechanically activated odontoblasts transmit sensory signals to adjacent odontoblasts and sensory neurons, initiating mechano-transduction and related downstream signaling pathways and eventually regulating the functional behaviors of odontoblasts in response to mechanical stimuli accordingly. (**E**) Odontoblasts convert external mechanical stimuli into biological signals and generate electrophysiological responses for sensory signaling transmission to adjacent TG neurons, facilitating mechano-sensing as well as nociception.

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
