# Peer review of "Dentin Mechanobiology: Bridging the Gap between Architecture and Function"

_ijms, 2024, doi:10.3390/ijms25115642_

Round 1

Reviewer 1 Report

Comments and Suggestions for Authors

The manuscript titled “Dentin Mechanobiology: Bridging the Gap between Architecture and Function” by Fu, X.; et al. is a Review work where the authors outlined the dental mechanosensing processes driven by external forces. Furthermore, the authors also discussed about future technology advances which could be implemented in this field. The most reventant outcomes shown in this Review work could be interesting for a specialized target audience. Furthermore, the manuscript is generally well-written.

However, it exists some points that need to be addressed (please, see them below detailed point-by-point). The most relevant outcomes remarked by the authors can contribute in the growth of many fields by the better understanding of the underlying mechanical response of  teeth components under certain conditions. For this reason, I will recommend the present scientific manuscript for further publication in the International Journal of Molecular Sciences once all the below described suggestions will be properly fixed.

Here, there exists some points that must be covered in order to improve the scientific quality of the manuscript paper:

1) KEYWORDS. The authors should consider to add the term “viscoelastic properties” in the keyword list.

2) “1. Clinical Implications of Dentin Mechanobiology in Dental Diseases” (lines 30-102). What are the global burden of dental diseases? Some quantitative insights should be provided in this regard (number of cases or economical impact). This will greatly benefit the potential readers to better understand the significance of this Review work.

3) “2. Dentin mechanosensing” (lines 103-324).

“Unlike enamel, dentin exhibits less brittleness and more viscoelasticity, provide the necessary flexibility (…) other harmful stimulati” (lines 112-114). Even if I agree with this statement provided by the authors it should be also mentioned how the local nanomechanics [1] of dentin display the neccesary performance to design smart adhesive-dentin interfaces. (This content could be also susceptible to be placed in the last manuscript version as Future Perspectives).

[1] Magazzù, A.; et al. Investigation of Soft Matter Nanomechanics by Atomic Force Microscopy and Optical Tweezers: A Comprehensive Review. Nanomaterials 2023, 13, 963. https://doi.org/10.3390/nano13060963.

[2] Yao, C.; et al. Nanoindentation Mapping and Bond Strength Study of Adhesive-Dentin Interfaces. Adv. Mater. Interfaces 2022, 9, 2101327. https://doi.org/10.1002/admi.202101327.

4) Then, some physical equations depicting how denting can sense the external forces and its subsequent behaviour (mechanical models) need to be added and discussed by the authors.

5) “3.1. Microchanneld platforms mimicking dentinal tubule structure” (lines 358-461). First, the grammatical issue “Microchanneld” from the subsection title should be fixed. Then, the authors mentioned the main advantages about the explotation of this technology. Is there any inherent limitations? In case affirmative, how they could be overcome? Could the platform material nature negatively impact on their future applications? Some discussion should be furnished in this regard.

6) CONCLUDING REMARKS AND FUTURE PERSPECTIVES. This section clearly states the most relevant outcomes found in this field and also the potential future action lines to pursue this research. No actions are requested from the authors.

Comments on the Quality of English Language

The manuscript is generally well-written. However, it may be desirable if the authors could recheck it in order to polish those final details susceptible to be improved.

Author Response

Response to Reviewer #1

We sincerely appreciate the thorough feedback given by the reviewer. Your comments have greatly enhanced our manuscript's quality and clarity. In response to your feedback, we have carefully addressed each concern raised, making necessary modifications accordingly. All the revisions have been marked with highlighted text in the revised version. Thank you again for your invaluable guidance in refining our manuscript.

Reviewer #1: The manuscript titled “Dentin Mechanobiology: Bridging the Gap between Architecture and Function” by Fu, X.; et al. is a Review work where the authors outlined the dental mechanosensing processes driven by external forces. Furthermore, the authors also discussed about future technology advances which could be implemented in this field. The most reventant outcomes shown in this Review work could be interesting for a specialized target audience. Furthermore, the manuscript is generally well-written.

However, it exists some points that need to be addressed (please, see them below detailed point-by-point). The most relevant outcomes remarked by the authors can contribute in the growth of many fields by the better understanding of the underlying mechanical response of  teeth components under certain conditions. For this reason, I will recommend the present scientific manuscript for further publication in the International Journal of Molecular Sciences once all the below described suggestions will be properly fixed.

Here, there exists some points that must be covered in order to improve the scientific quality of the manuscript paper:

  1. The authors should consider to add the term “viscoelastic properties” in the keyword list.
  • Thank you for the suggestion. We have included the term “viscoelastic properties” in the keyword list, as suggested (line 28).
  1. “1. Clinical Implications of Dentin Mechanobiology in Dental Diseases” (lines 30-102). What are the global burden of dental diseases? Some quantitative insights should be provided in this regard (number of cases or economical impact). This will greatly benefit the potential readers to better understand the significance of this Review work.
  • Thank you for your valuable feedback. We agree with your suggestion to provide quantitative insights into the global burden of dental diseases to enhance the understanding of the significance of our review work. In response, we have incorporated relevant quantitative information regarding the global burden of dental diseases in the revised manuscript (line 32).
  1. “2. Dentin mechanosensing” (lines 103-324).

“Unlike enamel, dentin exhibits less brittleness and more viscoelasticity, provide the necessary flexibility (…) other harmful stimulati” (lines 112-114). Even if I agree with this statement provided by the authors it should be also mentioned how the local nanomechanics [1] of dentin display the necessary performance to design smart adhesive-dentin interfaces. (This content could be also susceptible to be placed in the last manuscript version as Future Perspectives).

[1] Magazzù, A.; et al. Investigation of Soft Matter Nanomechanics by Atomic Force Microscopy and Optical Tweezers: A Comprehensive Review. Nanomaterials 2023, 13, 963. https://doi.org/10.3390/nano13060963.

[2] Yao, C.; et al. Nanoindentation Mapping and Bond Strength Study of Adhesive-Dentin Interfaces. Adv. Mater. Interfaces 2022, 9, 2101327. https://doi.org/10.1002/admi.202101327.

  • Thank you for your valuable input. We have incorporated a discussion on the local micro/nano-structures and -mechanics of dentin, as suggested by the reviewer, and referenced studies highlighting their importance for designing smart materials for successful dental regenerative therapy (Ref 152-154, line 718).
  1. Then, some physical equations depicting how dentin can sense the external forces and its subsequent behavior (mechanical models) need to be added and discussed by the authors.
  • Thank you for your suggestion. We agree that providing physical equations could enhance understanding of how dentin senses external forces. However, most studies have primarily focused on fluid mechanical modeling in dentinal microtubules through thermal sensing, with limited attention given to other mechanical forces. Comprehensive numerical computer simulations regarding mechanical models in dentin mechano-sensing are lacking, and existing models are based on the hydrodynamic theory, which has many controversies and ambiguities. Therefore, we have decided not to include physical equations in the manuscript. Instead, we emphasize the need for future research to address this gap. This has been discussed in the revised manuscript (line 672).
  1. “3.1. Microchanneld platforms mimicking dentinal tubule structure” (lines 358-461). First, the grammatical issue “Microchanneld” from the subsection title should be fixed. Then, the authors mentioned the main advantages about the explotation of this technology. Is there any inherent limitations? In case affirmative, how they could be overcome? Could the platform material nature negatively impact on their future applications? Some discussion should be furnished in this regard.
  • We have thoroughly revised the term to “Microchannel” in the revised manuscript.
  • Regarding the limitations of microchannel platforms, we acknowledge that current platforms only replicate the structure and morphology of dentin tubules, whereas, in vivo, dentin tubules develop concurrently with odontoblast differentiation from their progenitors. To address this limitation, future research should prioritize the development of platforms that stimulate the spontaneous formation of dentin tubules by odontoblasts, rather than replying on pre-formed guidance for odontoblast processing. Additionally, we have discussed the potential impact of microchannel platforms on their future applications (line 478).
  1. CONCLUDING REMARKS AND FUTURE PERSPECTIVES. This section clearly states the most relevant outcomes found in this field and also the potential future action lines to pursue this research. No actions are requested from the authors.
  • We are grateful for your feedback on the concluding remarks and future section of our manuscript. Following your suggestion, we have expanded the discussion. Thank you again for your comments; they have enhanced the clarity and comprehensiveness of this section.

Reviewer 2 Report

Comments and Suggestions for Authors

An interesting article, well written, but as a reviewer I have a few comments, mainly related to how individual articles in your work were selected.

The abstract

should include information about what databases were searched and how many articles were found describing this issue. Next, what conclusions can be drawn from this and what practical conclusion does your work have, why was it done?

Introduction

Line 35

This resilience suggests the presence of inherent mechanical adaptation mechanisms within the tooth structure to counter constant mechanical stress. - you mention a theory, it would be good to support it with some reference.

Line 57

and erosion [10], corrosion- what's the difference here? |Corrosion is a phenomenon resulting from the flow of electrostatic charges

It would be good to add 1-2 sentences about the treatment of hypersensitivity  it gives the answer to the main question of your publication about  dentin  nature.

Davari A, Ataei E, Assarzadeh H. Dentin hypersensitivity: etiology, diagnosis and treatment; a literature review. J Dent (Shiraz). 2013 Sep;14(3):136-45. PMID: 24724135; PMCID: PMC3927677.

Raszewski, Z.; Chojnacka, K.; Mikulewicz, M. Investigating Bioactive-Glass-Infused Gels for Enamel Remineralization: An In Vitro Study. J. Funct. Biomater. 2024, 15, 119. https://doi.org/10.3390/jfb15050119

 Line 135

. In tooth defects, dentin serves as the first defense barrier of the dentin-pulp complex against the invasion of exogenous stimuli-  enamel is the first protection layer.

Line 146

 Before starting the discussion about Mechanosensation it will be good to  add the definition of this term, please

Figure 1 is your or form  literature, if  from others do you have permission to publish this drawing?

Line 250-270

In every sentence you can use PIEZO1, you can replace it with other words receptors, conductive cells

Table 2, you need to separate the Gola text from the Finding text, because they merge into one and are difficult to read

As I wrote at the beginning, the article does not tell me what databases you used and in what years. What was the criterion for selecting articles to present your theory?

  What are the limitations of your tests?

There are many abbreviations used, so it would be good to present them again and explain what they mean at the end of the article. This makes it easier to read.

References

Chun, K.J.; Choi, H.; Lee, J.-Y. Comparison of mechanical property and role between enamel and dentin in the human teeth. J. Dent. Biomech 2014, 5 pages are missing

Good luck with your further research!

Author Response

Response to Reviewer #2

We sincerely appreciate the reviewer’s insightful and crucial comments on our manuscript. Your input has greatly contributed to enhancing the quality and clarity of our manuscript. We have taken your comments seriously, in response to your feedback, we have carefully addressed the concerns raised in a point-by-point manner and made necessary modifications to the manuscript accordingly. All the revisions have been marked with highlighted text in the revised version. Thank you once again for your invaluable guidance in refining our manuscript.

Reviewer #2: An interesting article, well written, but as a reviewer I have a few comments, mainly related to how individual articles in your work were selected.

  1. The abstract

should include information about what databases were searched and how many articles were found describing this issue. Next, what conclusions can be drawn from this and what practical conclusion does your work have, why was it done?

  • Thank you for your valuable feedback. We conducted searches on articles related to dentin mechanobiology, dentin diseases, dentin-mimicking platforms, and materials in PubMed (3,251 results), ScienceDirect, (36,900 results), and Google Scholar (544,000 results). From these databases, we selected the most relevant articles (~150 references) for inclusion in our review.
  • Furthermore, in PubMed, from 2000 to 2024, we observed a gradual increase in articles using the keyword “dentin mechanobiology” (See below). However, the accumulated number of articles was only 101, indicating that this topic has not received significant attention despite its importance.
  • Additionally, we have incorporated quantitative data on the global burden of dental diseases and discussed the importance of understanding dentin mechanobiology in the revised manuscript (line 32).

Introduction

  1. Line 35

This resilience suggests the presence of inherent mechanical adaptation mechanisms within the tooth structure to counter constant mechanical stress. - you mention a theory, it would be good to support it with some reference.

  • Thank you for your suggestion. As you recommended, we have included references (Ref 3-8) to support the suggestion regarding inherent mechanical adaptation mechanisms within the tooth structure (line 53).
  1. Line 57

and erosion [10], corrosion- what's the difference here? |Corrosion is a phenomenon resulting from the flow of electrostatic charges

  • Thank you for your valuable feedback. We have deleted the word “corrosion” as it represents one of the contributing factors to the erosion of the tooth surface (line 76).
  1. It would be good to add 1-2 sentences about the treatment of hypersensitivity it gives the answer to the main question of your publication about dentin nature.

Davari A, Ataei E, Assarzadeh H. Dentin hypersensitivity: etiology, diagnosis and treatment; a literature review. J Dent (Shiraz). 2013 Sep;14(3):136-45. PMID: 24724135; PMCID: PMC3927677.

Raszewski, Z.; Chojnacka, K.; Mikulewicz, M. Investigating Bioactive-Glass-Infused Gels for Enamel Remineralization: An In Vitro Study. J. Funct. Biomater. 2024, 15, 119. https://doi.org/10.3390/jfb15050119

  • Thank you for your suggestion. We have incorporated additional references (Ref 26 and 27) addressing the clinical management of dentin hypersensitivity (lines 92).
  1. Line 135

In tooth defects, dentin serves as the first defense barrier of the dentin-pulp complex against the invasion of exogenous stimuli- enamel is the first protection layer.

  • Thank you for your clarification. We have revised the sentence to more accurately reflect the focus on the protective function of dentin for the dentin-pulp complex. The revised sentence now reads, “In tooth defects, dentin serves as a protective barrier for the dentin-pulp complex, defending against the invasion of external stimuli.” (line 161).
  1. Line 146

Before starting the discussion about Mechanosensation it will be good to add the definition of this term, please

  • Thank you for your suggestion. We have incorporated the definition of mechano-sensation and mechano-transduction in the revised manuscript (line 172).
  1. Figure 1 is your or form literature, if from others do you have permission to publish this drawing?
  • Thank you for addressing this concern. Figure 1 is an original illustration created by us.
  1. Line 250-270

In every sentence you can use PIEZO1, you can replace it with other words receptors, conductive cells

  • Thank you for your comment. PIEZO channels are mechano-sensitive ion channels, and their specific role is crucial in the context of our manuscript. Therefore, we have chosen to maintain the repeated use of “PIEZO” instead of replacing it with other terms such as receptors.
  1. Table 2, you need to separate the Goal text from the Finding text, because they merge into one and are difficult to read
  • Thank you for bringing this to our attention. We have inserted additional spacing between columns to enhance the readability of the table (Table 2).
  1. As I wrote at the beginning, the article does not tell me what databases you used and in what years. What was the criterion for selecting articles to present your theory? What are the limitations of your tests?
  • Thank you for your inquiry. We conducted searches on three databases, namely PubMed, Science Direct, and Google Scholar, resulting in a total of 584,151 articles. From these, we selected the most relevant articles (~150 references), focusing particularly on those published within the last five years.
  • Regarding the limitations of our study, we acknowledge the scarcity of research articles delving deeply into dentin mechanobiology. Additionally, existing materials and platforms predominantly emphasize mimicking chemical properties rather than physical features. Therefore, our discussion is based on addressing potential rather than actual mechanobiological findings. We have clarified these limitations in the revised manuscript (line 672).
  1. There are many abbreviations used, so it would be good to present them again and explain what they mean at the end of the article. This makes it easier to read.
  • We appreciate your concern regarding the use of abbreviations. To enhance clarity, we have made sure to provide the full name of each abbreviation when it is first mentioned in the manuscript and in a footnote accompanying the tables. This approach should make it easier for readers to understand and follow along.
  1. References

Chun, K.J.; Choi, H.; Lee, J.-Y. Comparison of mechanical property and role between enamel and dentin in the human teeth. J. Dent. Biomech 2014, 5 pages are missing

  • Thank you for bringing this to our attention. We have revised the reference information to ensure its accuracy (Ref 10).

Reviewer 3 Report

Comments and Suggestions for Authors

Dear authors,

Thank you for your well structured review article. There are minor revisions necessary. Please also include information upon collagen types present and their role in bacterial adhesion. Characterize which role dentinal tubes play in bacterial infection of dentin. For this purpose, it is advised to also include doi: 10.1016/j.dental.2014.03.003.

Author Response

Response to Reviewer #3

We genuinely appreciate the reviewer's thoughtful evaluation and valuable feedback on our manuscript. Your insights have significantly enhanced the clarity and depth of our work. Taking your suggestions seriously, we have made revisions to address the concerns raised. All revisions have been highlighted in the revised version. Thank you once again for your invaluable guidance in refining our manuscript.

Reviewer #3:

Dear authors,

Thank you for your well structured review article. There are minor revisions necessary.

  1. Please also include information upon collagen types present and their role in bacterial adhesion. Characterize which role dentinal tubes play in bacterial infection of dentin. For this purpose, it is advised to also include doi: 10.1016/j.dental.2014.03.003.
  • Thank you for your suggestion. We have incorporated discussions on the role of type I collagen fibrils in bacterial invasion (Ref 37, line 141) and referenced the article that discusses the impact of the diameter of dentin microtubules on bacterial colonization (Ref 107, line 422).

Round 2

Reviewer 1 Report

Comments and Suggestions for Authors

All the suggestions raised by the Reviewers were successfully addressed by the authors and subsequently the scientific quality of the manuscript revised version was greatly improved. For all the aforementioned aspects, I warmly endorse this research for further publication in the International Journal of Molecular Sciences.